# Association between hypertriglyceridemic waist phenotype and hypogonadism in Taiwanese adult men

Sheng-Kuang Wang[1,2], Meng-Chih Lee[3,4,5,6], Chia-Lien Hung[7], Hsin-Hung Chen[8,9,10,11], Chun-Cheng Liao[7,12,13]*, Yu-Lung Chiu[14,15]*

1 Graduate Institute of Medical Sciences, National Defense Medical Center, Taipei, Taiwan, 2 Medical Administrative Department, Taichung Army Forces General Hospital, Taichung, Taiwan, 3 Department of Family Medicine, Taichung Hospital, Ministry of Health and Welfare, Taichung, Taiwan, 4 College of Management, Chaoyang University of Technology, Taichung, Taiwan, 5 Institute of Medicine, Chung Shan Medical University, Taichung, Taiwan, 6 Institute of Population Health Sciences, National Health Research Institutes, Miaoli, Taiwan, 7 Department of Medical Education and Research, Taichung Armed Forces General Hospital, Taichung, Taiwan, 8 Division of Endocrinology and Metabolism, Department of Internal Medicine, Asia University Hospital, Taichung, Taiwan, 9 School of Medicine, Chung Shan Medical University, Taichung, Taiwan, 10 Department of Law, Providence University, Taichung, Taiwan, 11 Chung Sheng clinic, Nantou, Taiwan, 12 Department of Family Medicine, Taichung Armed Forces General Hospital, Taichung, Taiwan, 13 School of Medicine, National Defense Medical Center, Taipei, Taiwan, 14 Graduate Institute of Life Sciences, National Defense Medical Center, Taipei, Taiwan, 15 School of Public Health, National Defense Medical Center, Taipei, Taiwan

* milkbottle97@yahoo.com.tw (CCL); long_ruth0624@mail.ndmctsgh.edu.tw (YLC)

## Abstract

### Background

Aging-related hypogonadism in men is related to the deterioration of overall health. Those with this disease rarely receive treatment. The hypertriglyceridemic waist (HTGW) phenotype is a tool for predicting abnormalities of cardiovascular metabolism. However, the relationship between the HTGW phenotype and hypogonadism remains undetermined. This study aimed to determine the association between HTGW phenotype and hypogonadism in different age groups.

### Methods

Data of this cross-sectional study were obtained from MJ Health Screening Center in Taiwan from 2007 to 2016. The HTGW phenotype was divided into four categories based on whether the waist circumference (WC) and triglyceride levels were normal. WC of <90 cm and triglyceride level of <150 mg/dL were defined as normal. Hypogonadism was defined as a testosterone level of <300 ng/dL.

### Results

Overall, 6442 male participants were divided into three age groups: <50, 50–64, and ≥65 years (n = 4135, 1958, and 349; age groups 1, 2, and 3, respectively). The overall prevalence of hypogonadism was 10.6%. In group 1, participants with HTGW (odds ratio, 1.98; 95% confidence interval (CI), 1.354–2.896) had a higher risk of hypogonadism than those

**Data Availability Statement:** The data used in this project is owned by a third-party organization called the MJ Health Screening Center, which collects self-paid health examination of its

members. We had to sign strict confidentiality and data protection agreements to be allowed to work with the data. Consequently, we are not allowed to share a minimal dataset publicly. However, the data we used is also available by the MJ Health Research Foundation (contact via contact_us@mjhrf.org) for researchers who meet the criteria for access to the data.

**Funding:** The study was supported by the Taichung Armed Forces General Hospital (grant numbers: MJHRF2021003A) and the funder had no role in study design, data collection and analysis, decision to publish, or preparation of the manuscript.

**Competing interests:** The authors have declared that no competing interests exist.

with normal WC and normal triglyceride levels after adjustment for body mass index and fasting blood glucose level. In group 2, participants with HTGW (odds ratio, 1.873; 95% CI, 1.099–3.193) had an increased risk of hypogonadism after adjustment for body mass index, fasting blood glucose level, Cholesterol levels, high-density lipoprotein (HDL) levels, low-density lipoprotein (LDL) levels and smoking status. However, no relationship was observed between HTGW phenotype and hypogonadism in group 3.

## Conclusion

HTGW phenotype was highly associated with hypogonadism in Taiwanese adult men. More attention should be paid to men aged <50 years with HTGW.

## Introduction

Male hypogonadism is characterized by a deficiency of testosterone, a critical hormone required for sexual, cognitive, and body functions and development [1]. Studies have found that testosterone deficiency not only affects the quality of life but also increases the risk of erectile dysfunction, metabolic syndrome, diabetes, osteoporosis, fracture, and cardiovascular disease [2–7]. Studies have also reported that middle-aged and elderly men with low testosterone levels have increased overall mortality and cardiovascular disease mortality rates in long-term follow-up [8–12]. Therefore, low testosterone levels may be regarded as a threat to men's health. Based on population studies, the prevalence of hypogonadism is 3.8%–28.1%. Hypogonadism is estimated to affect 4 million men in the United States; low testosterone levels are especially common in older men. However, the treatment rates varied in different populations and were generally low (9.65%–11.3% of men with hypogonadism) [13, 14]. In addition, many men with hypogonadism who may benefit from testosterone replacement are not receiving such treatment [15].

The hypertriglyceridemic waist (HTGW) phenotype was proposed in 2000, which was defined as high waist circumference (WC) and triglyceride (TG) levels [16]. Many studies indicated that the HTGW phenotype is associated with diabetes and cardiovascular disease and is a simple and inexpensive tool for predicting cardiovascular metabolism abnormalities in people with excess visceral adipose tissue [17–20]. Both HTGW phenotype and hypogonadism are associated with diseases such as diabetes and cardiovascular disease, and HTGW phenotype is an easier and more economical screening tool. However, studies have rarely discussed the relationship between hypogonadism and HTGW, even in Taiwan. Therefore, we conducted this cross-sectional study to examine the relationship between hypogonadism and HTGW in middle-aged men in Taiwan.

## Materials and methods

### Study population

This cross-sectional study was conducted from 2007 to 2016. The participant flow chart is shown in Fig 1. Data were retrieved from the MJ Health Screening Center in Taiwan, a large private health examination institute. This institute provides self-paid health examination services in Taiwan at a general cost of approximately 200–730 USD. The serum testosterone test costs approximately 17 USD.

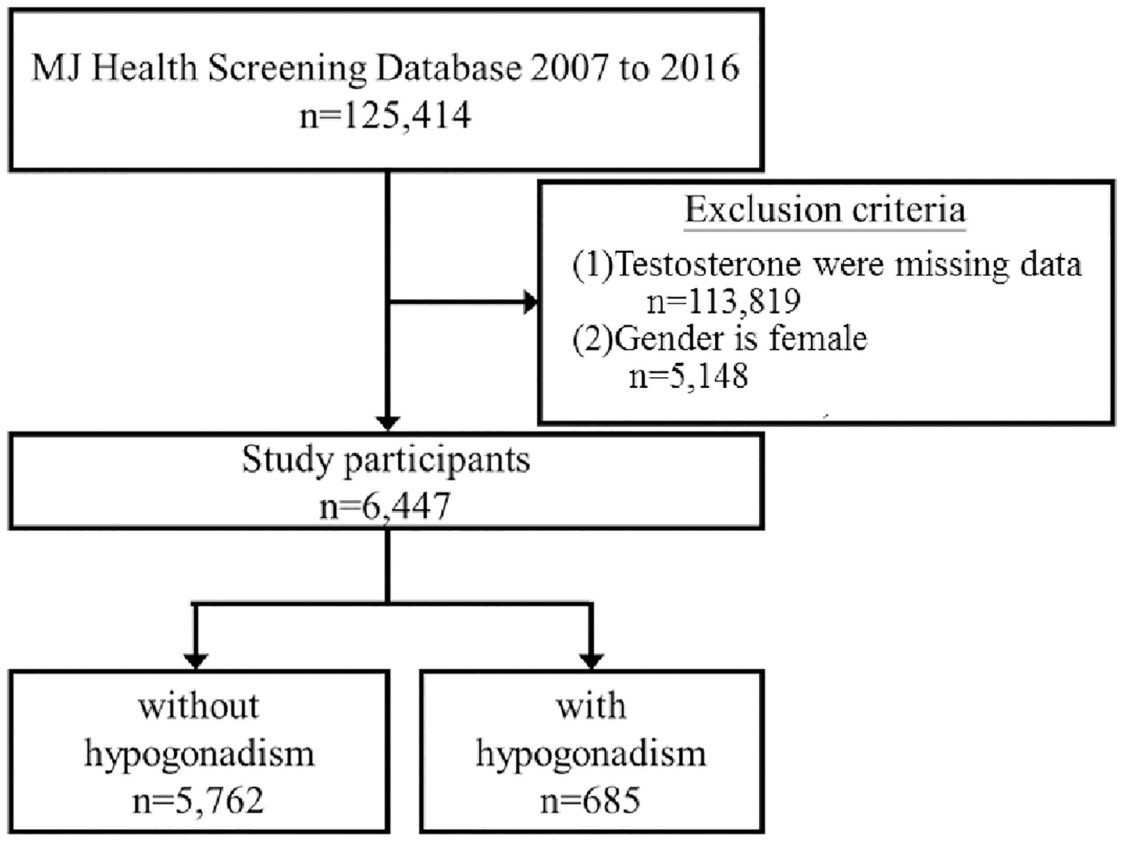

**Fig 1. Flow diagram of participant enrollment.**

A total of 125,414 male participants were originally included in the study. Subsequently, we excluded participants with prostate cancer and insufficient data on serum testosterone level. We categorized the participants into three groups according to age ($<$50, 50–64, and $\geq$65 years, with n = 4135, 1958, and 349 participants, respectively) and further classified them into those with and those without hypogonadism.

Prior to the health examination, all the participants provided written informed consent to participate in the study. The MJ Health Research Foundation authorized (MJHRF2019016A) the use of all the data they provided for this study. All identifiable participant data were removed to maintain participant anonymity. The data collection procedure and detailed characteristics of the study population are reported elsewhere [5]. The research protocol was approved by the Institutional Review Board of the Tri-Service General Hospital, National Defense Medical Center (A202005160).

### Definition of the participants' baseline characteristics

Data on the participants' body mass index (BMI; kg/m$^2$), WC, TG level, fasting blood glucose (FBG) level, and testosterone level were retrieved. WC was measured at the middle level of the top of the hip bone and bottom of the rib bone without clothing that might interfere with the measurement. The homogeneous direct method (Toshiba C8000) for measurement of high-density lipoprotein (HDL)-cholesterol, GPO-POD-ESPT method (Toshiba C8000) for measurement of TG, and HK.G-6-PD.NADP method (Toshiba C8000) for measurement of FBG

were employed. A chemiluminescent micro-particle immunoassay (ARCHITECT i2000) was performed to measure the testosterone level. The details of these parameters have been reported previously [5].

### 2.3. Definitions of hyertriglyceridemic waist and hypogonadism

Four phenotypes were categorized in accordance with the 2005 American Heart Association/National Heart, Lung, and Blood Institute guideline for metabolic syndrome [21]. WC < 90 cm for men and TG level < 150 mg/dL were defined as normal. The four phenotypes [16] were as follows: (1) normal waist and normal TG level (NWNT); (2) hypertriglyceridemia and normal waist (HTG); (3) enlarged waist and normal TG level (EW); and (4) WC ≥ 90 cm and serum TG level ≥ 150 mg/dL in men (HTGW). Hypogonadism was defined as total testosterone level of < 300 ng/dL for men [22, 23].

### 2.4. Statistical analyses

An independent t-test was performed to compare age, BMI, WC, TG level, and FBG level between the participants with and without hypogonadism. The TG level and WC (four phenotypes) of participants with and without hypogonadism were compared using a chi-square test. Finally, a logistic regression model was applied to analyze the relationships between those with or without hypogonadism and the four phenotypes after adjusting for BMI, FBG levels, cholesterol levels, HDL levels, LDL levels and smoking status in the different age groups. All statistical analyses were conducted using SPSS version 22.0 (IBM, Armonk, NY, USA) software, and a p-value of <0.05 was considered statistically significant.

## Results

### Baseline characteristics of the participants

A total of 6,442 men aged <50 years (n = 4135, age group 1), 50–64 years (n = 1958, age group 2), and ≥65 years (n = 349, age group 3) participated in the study. In age groups 1 and 2, the mean BMI, WC, and TG and FBG levels were significantly higher in the hypogonadism group than in the non-hypogonadism group (p < 0.05; Table 1). The HDL level was significantly lower in the hypogonadism group than in the non-hypogonadism group. Furthermore, the cholesterol and LDL levels were significantly higher in the hypogonadism group than in the non-hypogonadism group (age group1, p<0.05; Table 1). There were no statistically significant differences in FBG, cholesterol, or LDL levels in age group 3 between the hypogonadism and non-hypogonadism groups (p >0.05; Table 1). Participants with hypogonadism were significantly more likely to be non-smokers (p<0.001).

### Correlations between phenotypes and hypogonadism

The overall prevalence of hypogonadism among all participants was 10.63%. As shown in Table 2, the distribution of phenotypes (NWNT, EW, HTG, and HTGW) was significantly different between participants with and without hypogonadism (p<0.001). Participants without hypogonadism were more likely to have the NWNT phenotype than those with hypobonadism.

The prevalence of each phenotype in those with and without hypogonadism in each age group is shown in Table 3. In age group 1, participants with hypogonadism were significantly more likely to be HTGW than those without (p<0.001). In age group 2, the NWNT was significantly more prevalent among those with hypogonadism than those without (p<0.001).

**Table 1. Characteristics of participants in different age groups with and without hypogonadism.**

| | Age < 50 years (n = 4135) | | | Age 50–64 years (n = 1958) | | | Age ≥ 65 years (n = 349) | | |
|---|---|---|---|---|---|---|---|---|---|
| | Without hypogonadis m (n = 3696) | With hypogonadism (n = 439) | p | Without hypogonadism (n = 1742) | With hypogonadism (n = 216) | p | Without hypogonadis m (n = 319) | With hypogonadis m (n = 30) | p-value |
| Age | 39.0 ± 6.7 | 39.4 ± 5.9 | 0.166 | 55.5 ± 4.1 | 55.3 ± 4.2 | 0.498 | 70.0 ± 4.9 | 70.4 ± 4.2 | 0.660 |
| BMI | 24.6 ± 3.4 | 27.9 ± 4.3 | <0.001[***] | 24.6 ± 2.9 | 26.8 ± 3.4 | <0.001[***] | 24.4 ± 2.9 | 26.3 ± 3.1 | <0.001[**] |
| WC | 83.2 ± 8.7 | 90.9 ± 9.5 | <0.001[***] | 84.5 ± 7.9 | 90.0 ± 8.8 | <0.001[***] | 86.2 ± 8.3 | 91.5 ± 8.0 | <0.001[**] |
| TG | 136.0 ± 105.5 | 186.7 ± 154.2 | <0.001[***] | 133.3 ± 92.3 | 167.8 ± 91.0 | <0.001[***] | 111.8 ± 72.8 | 148.6 ± 113.0 | 0.036[*] |
| FBG | 102.2 ± 16.7 | 110.9 ± 31.3 | <0.001[***] | 109.6 ± 21.3 | 118.8 ± 27.8 | <0.001[***] | 113.1 ± 25.0 | 117.2 ± 20.2 | 0.381 |
| cholesterol | 198.44±34.37 | 203.36±35.67 | 0.005[**] | 200.78±34.66 | 199.75±37.77 | 0.686 | 193.30±34.72 | 180.77±28.29 | 0.056 |
| HDL | 51.50±11.03 | 47.07±9.48 | <0.001[***] | 53.12±12.32 | 47.10±9.67 | <0.001[***] | 53.84±11.97 | 45.90±9.01 | <0.001[***] |
| LDL | 122.78±32.06 | 127.97±32.52 | 0.001[**] | 122.80±31.56 | 124.27±32.33 | 0.518 | 117.79±32.91 | 115.50±24.83 | 0.711 |
| T | 542.3 ± 188.1 | 241.6 ± 52.0 | <0.001[***] | 553.6 ± 224.6 | 237.6 ± 53.1 | <0.001[***] | 570.7 ± 236.1 | 216.1 ± 74.6 | <0.001[***] |
| Smoking status | | | 0.007[**] | | | 0.011[*] | | | 0.829 |
| Non-smoker | 2,050(56.9) | 270(63.4) | | 933(55.6) | 134(64.1) | | 208(67.1) | 20(71.4) | |
| Past smoker | 805(22.4) | 94(22.1) | | 438(26.1) | 53(25.4) | | 71(22.9) | 5(17.9) | |
| Smoker | 746(20.7) | 62(14.6) | | 308(18.3) | 22(10.5) | | 31(10.0) | 3(10.7) | |

BMI: body mass index (kg/m$^2$); WC: waist circumference (cm); TG: triglyceride (mg/dL); FBG: fasting blood glucose (mg/dL); cholesterol, Total Cholesterol (mg/dL); LDL, low-density lipoprotein (mg/dL); HDL, high-density lipoprotein (mg/dL); T: testosterone (ng/dL).

[*] p < 0.05;

[**] p < 0.01;

[***] p < 0.001.

However, there were no significant differences in phenotypes between those with and without hypogonadismin age group 3.

## Risk of hypogonadism according to the four phenotypes

As shown in Fig 2, we investigated the effects of the four phenotypes on the risk of hypogonadism. In age group 1, participants with the HTGW (odds ratio [OR], 1.980; 95% confidence interval [CI], 1.354–2.896) and HTG phenotype (OR, 1.803; 95% CI, 1.297–2.505) had higher risks of hypogonadism than those with the NWNT phenotype, after adjustment for BMI, FBG levels, cholesterol levels, HDL levels, LDL levels and smoking status. In age group 2, participants with the HTGW (OR, 1.873; 95% CI, 1.099–3.193) and HTG (OR, 2.098; 95% CI, 1.334–

**Table 2. Incidence of hypogonadism in the four phenotypes.**

| | Without hypogonadism (n = 5762) | With hypogonadism (n = 685) | p-value |
|---|---|---|---|
| **Prevalence** | | | |
| Overall | | **10.63%** | |
| **Phenotype** | | | <0.001 [***] |
| NWNT | 3,357 (58.3) | 194 (28.3) | |
| EW | 734 (12.7) | 150 (21.9) | |
| HTG | 1,068 (18.5) | 150 (21.9) | |
| HTGW | 603 (10.5) | 191 (27.9) | |

**Table 3. Incidence of the four phenotypes in the different age groups with and without hypogonadism.**

| | Age < 50 years (n = 4135, age group 1) | | | Age 50–64 years (n = 1958, age group 2) | | | Age ≥ 65 years (n = 349, age group 3) | | |
|---|---|---|---|---|---|---|---|---|---|
| | Without hypogonadism (n = 3696) | With hypogonadism (n = 439) | p | Without hypogonadism (n = 1742) | With hypogonadism (n = 216) | p | Without hypogonadism (n = 319) | With hypogonadism (n = 30) | p-value |
| Prevalence | | | | | | | | | |
| Age group | | 439(10.62) | | | 216(11.03) | | | 30(8.60) | |
| Phenotype | | | <0.001 *** | | | <0.001 *** | | | 0.071 |
| NWNT | 2,209 (59.8) | 124 (28.2) | | 972 (55.8) | 60 (27.8) | | 173 (54.2) | 10 (33.3) | |
| EW | 406 (11.0) | 85 (19.4) | | 264 (15.2) | 54 (25.0) | | 64 (20.1) | 11 (36.7) | |
| HTG | 709 (19.2) | 97 (22.1) | | 318 (18.3) | 50 (23.1) | | 41 (12.9) | 3 (10.0) | |
| HTGW | 372 (10.1) | 133 (30.3) | | 188 (10.8) | 52 (24.1) | | 41 (12.9) | 6 (20.0) | |

* p < 0.05;

** p < 0.01;

*** p < 0.001.

3.298) phenotypes also had higher risks of hypogonadism than those with NWNT after adjustment for BMI, FBG levels, cholesterol levels, HDL levels, LDL levels and smoking status. However, no relationship was observed between the phenotypes and hypogonadism in age group 3.

NWNT, normal waist circumference and normal triglyceride levels; HTG, hypertriglyceridemia and normal waist circumference; EW, enlarged waist and normal triglyceride levels; HTGW, hypertriglyceridemia and waist (circumference ≥ 90 cm).

## Discussion

This is the first study to examine the correlation between the HTGW phenotype and hypogonadism since the concept of the HTGW phenotype as a relatively inexpensive and common screening tool in 2000 [16]. Our research shows that the HTGW phenotype is independently associated with hypogonadism in Taiwanese adult men aged <65 years. Our findings suggest that the management of high TG and WC may have potential health benefits for the treatment of hypogonadism. Moreover, in the stratified analysis of age groups, we found that the HTGW group in the middle-aged population (age < 50 years) had the highest OR (1.980) for hypogonadism. This implies that the HTGW phenotype as a screening tool for hypogonadism might serve as a basis for early intervention in the middle-aged population, which comprises younger adults and the main workforce, to reduce the incidence of hypogonadism and its more serious complications.

Several studies have shown that visceral obesity in men is associated with low testosterone levels, male infertility, and erectile dysfunction [24–28]. In animal experiments, weight loss was associated with a rise in testosterone, free testosterone (FT) and sex hormone-binding globulin (SHBG), whereas weight gain was associated with a fall in these levels [29]. However, hypogonadism may induce visceral obesity. In castrated animal model, weight significantly increased, especially in terms of visceral fat accumulation [30]. This implies that hypogonadism has a bidirectional relationship with obesity. Among the components of metabolic syndrome, it has the strongest association with visceral obesity [31, 32]. Indeed, male hypogonadism can result from complex interactions between lifestyle, metabolic health and genetic background. Endocrine disruption caused by metabolic diseases can trigger the onset of hypogonadism, although the underlying mechanisms are not fully understood [33]. Metabolic disorders may modify the hypothalamic pituitary gonadal (HPA) axis by periodically

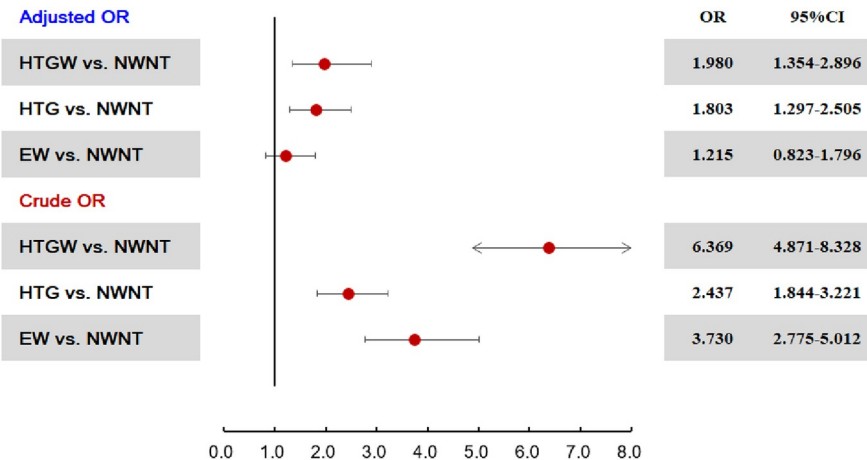

Odds ratio for hypogonadism in age group 1 (aged <50 years)

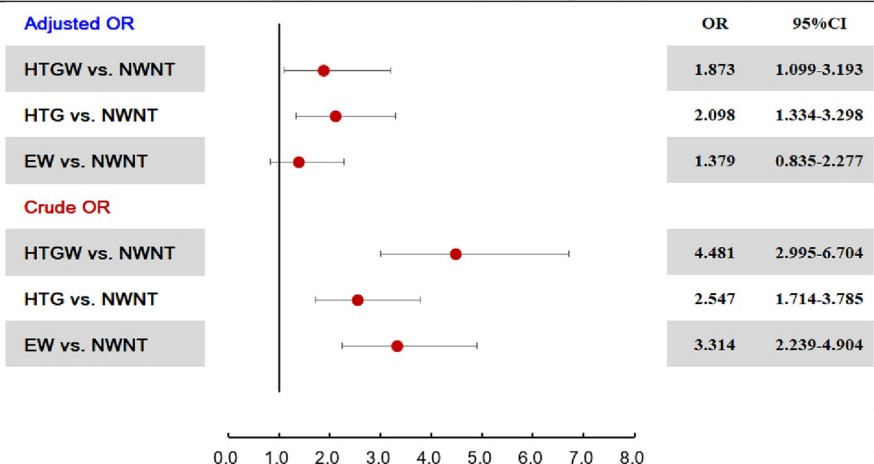

Odds ratio for hypogonadism in age group 2 (aged 50-64 years)

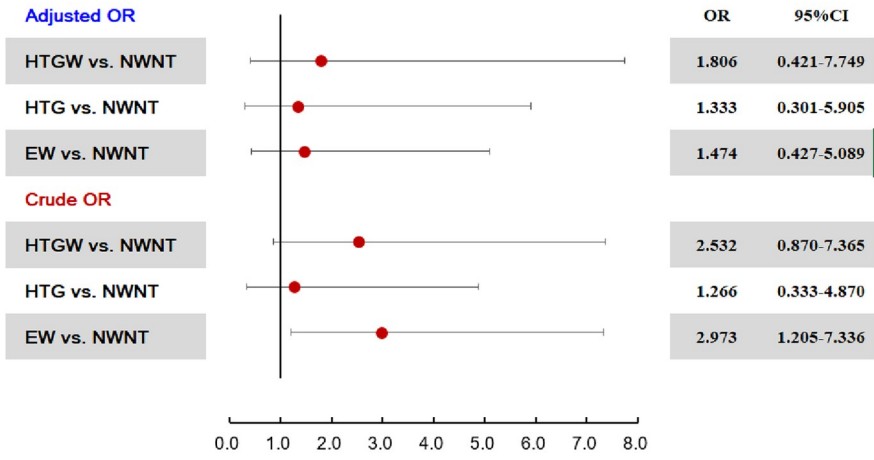

Odds ratio for hypogonadism in age group 3 (aged ≥65 years)

**Fig 2. Odds ratio for hypogonadism stratified by age group.** Adjusted OR were adjusted for body mass index, fasting blood glucose levels, cholesterol levels, high-density lipoprotein levels, low-density lipoprotein levels, and smoking status.

suppressing one or more of its components, whereas, in permanent forms of hypogonadism one or more of the HPA axis components has irrevocably loses functionality [34].

Instead of measuring BMI alone, the HTGW phenotype represents a simple screening method to identify individuals likely to have excess visceral adiposity and ectopic fat [24]. The TG level and WC are commonly measured in health examinations in Taiwan, but testosterone level is rarely measured in middle-aged people. Therefore, by screening of the HTGW phenotype, a surrogate marker of visceral obesity that combines at-risk TG level measurement with at-risk WC measurement has been proposed [16, 17]. Screening of the HTGW phenotype allows for earlier identification of the high-risk group of patients with hypogonadism and further examination of testosterone level in this population to facilitate early diagnosis and treatment. However, in current geriatric practice, testosterone analysis during routine health examinations is yet to be included. Therefore, the present study suggests incorporating testosterone analysis in the health examination list for men, especially those aged <50 years with HTGW. This may be used as a more economical and effective screening method for hypogonadism to prevent the subsequent development of hypogonadism and related serious complications.

Whether the serum testosterone level in men decreases with age is still disputable. Although the average testosterone level was found to decline with age in many studies [23, 35, 36], in the present study, the prevalence of hypogonadism in each age group did not increase with age (<50 years: 10.62%; $50 \leq age < 65$ years: 11.03%; and age $\geq$ 65 years: 8.60%). This age-related decline in testosterone level was not observed in studies performed in China [37, 38] and Japan [39]. The precise causes of this variation are unknown, but ethnic factors or differences in study design may account for this variation. As the health examinations performed at the MJ Health Screening Center are self-paid, our research participants might have had deviations in their income level or health awareness.

This study has several limitations. First, this cross-sectional study could not clarify the causality between the HTGW phenotype and hypogonadism in the Asian population. Further studies with a prospective design are necessary to verify our findings in different ethnic populations. Second, we evaluated testosterone deficiency based on a single measurement of total testosterone level because the FT data could not be obtained from the health examination database used in this study. Furthermore, some pathological conditions that may affect testes function (eg injury, trauma, infection or tumor, endocrine disease, etc.) were not able to include in our data. We hope to conduct another retrospective study in the future to investigate the relationship between the HTGW phenotype and FT without previous pathological conditions. Third, this study did not use imaging technologies such as computed tomography and magnetic resonance imaging to evaluate the visceral fat content of the participants, so we could not understand the relationship between visceral fat and testosterone level. Additional studies are needed to examine this relationship.

## Conclusions

Based on the results of nearly 6447 middle-aged and elderly participants, the present study shows that the HTGW phenotype was highly associated with hypogonadism in Taiwanese adult men. More attention should be paid to people aged <50 years who have the HTGW phenotype to reduce the serious complications of hypogonadism.

## Acknowledgments

This study received assistance by the Department of Medical Education and Research, Taichung Armed Forces General Hospital. All or part of the data used in this research were

authorized by and received from MJ Health Research Foundation (Authorization Code: MJHRF2021003A). Any interpretation or conclusion described in this paper does not represent the views of MJ Health Research Foundation.

## Author Contributions

**Conceptualization:** Sheng-Kuang Wang, Meng-Chih Lee, Hsin-Hung Chen, Chun-Cheng Liao, Yu-Lung Chiu.

**Data curation:** Sheng-Kuang Wang, Chia-Lien Hung, Chun-Cheng Liao, Yu-Lung Chiu.

**Funding acquisition:** Sheng-Kuang Wang.

**Methodology:** Sheng-Kuang Wang.

**Writing – original draft:** Sheng-Kuang Wang, Chun-Cheng Liao, Yu-Lung Chiu.

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
