## [Decision Letter · Decision Letter 0]

24 Nov 2021

PONE-D-21-22955Association between hypertriglyceridemic waist phenotype and hypogonadism in Taiwanese adult menPLOS ONE

Dear Dr. Chiu,

Thank you for submitting your manuscript to PLOS ONE. After careful consideration, we feel that it has merit but does not fully meet PLOS ONE’s publication criteria as it currently stands. Therefore, we invite you to submit a revised version of the manuscript that addresses the points raised during the review process.

We look forward to receiving your revised manuscript.

Kind regards,

Giacomo Pucci

Academic Editor

PLOS ONE

Journal Requirements:

“This study received assistance by the Department of Medical Education and Research, Taichung Armed Forces General Hospital. All or part of the data used in this research were authorized by and received from MJ Health Research Foundation (Authorization Code: MJHRF2021003A). Any interpretation or conclusion described in this paper does not represent the views of MJ Health Research Foundation.”

Reviewers' comments:

Reviewer's Responses to Questions

**Comments to the Author**

1. Is the manuscript technically sound, and do the data support the conclusions?

Reviewer #1: No

Reviewer #2: Partly

2. Has the statistical analysis been performed appropriately and rigorously? 

Reviewer #1: No

Reviewer #2: Yes

3. Have the authors made all data underlying the findings in their manuscript fully available?

Reviewer #1: Yes

Reviewer #2: No

4. Is the manuscript presented in an intelligible fashion and written in standard English?

Reviewer #1: No

Reviewer #2: Yes

5. Review Comments to the Author

Reviewer #1: In the manuscript submitted by Sheng-Kuang Wang et al. about “Association between hypertriglyceridemic waist phenotype and hypogonadism in Taiwanese adult men”, authors aim to investigate the potential relationship between hypogonadism (defined as plasma testosterone level <300 ng/dL) and the hypertriglyceridemic waist phenotype (HTGW, defined as waist circumference >90 cm and plasma triglycerides >150 mg/dL) in a large population of Taiwanese males. Authors divided the population in different categories according to age, hypogonadism (yes or no) and the waist circumference-triglycerides phenotype (NWNT; HTG; EW; HTGW), so they performed chi-square test and logistic regression analyses. They conclude that (lines 151-153) “Our research shows that the HTGW phenotype is independently associated with hypogonadism in Taiwanese adult men aged <65 years. Our findings suggest that the management of high TG and WC may have potential health benefits for the treatment of hypogonadism”

Results are reported in an ambiguous and confusing manner and these conclusions are not supported by the data submitted and the methodology used.

1. lines 118-120 is not clear what the p value is referring to. In the text authors report that “there was a statistically significant difference between the phenotypes (NWNT, EW, HTG, and HTGW) and hypogonadism (p < 0.001).”, but from table 2 p value seems about the difference between group with and without hypogonadism. However, it appears from numbers displayed in table 2 that there is a higher prevalence of HTGW and a lower prevalence of NWNT in those with hypogonadism than in those without hypogonadism, but among those with hypogonadism it seems that the prevalence of HTGW is similar to that of NWNT. Nevertheless these are only impressions from numbers because pairwise p values are not reported. The same for the analyses showed in table 3 and lines 126-132.

2. Authors derives from chi-square test the correlation between these subgroups, but chi-square actually report information about the difference in distribution and not about correlation. Consequently, the logistic regression analyses is not really supported from previous analyses and implies a causal relationship that is not justified by the presented data.

I suggest analysing the variables as continuous variables and using appropriate tests to highlight possible correlations and then detect the presence of potential hypogonadism independent determinants through multivariate regression analysis.

Reviewer #2: Wang and colleagues present a cross-sectional study in which the association between hypertriglyceridemic waist (HTGW: defined as waist circumference >90 cm and plasma triglycerides >150 mg/dL) phenotype and hypogonadism (i.e., plasma testosterone level <300 ng/dL) has been analyzed in three different groups, based on age, of adult male Taiwanese. Their analysis shows a significant impact of HTGW on hypogonadism, especially in patients under 65 years, which present a higher risk of low testosterone levels.

Despite this topic is very relevant and attractive, I found the present study a bit confused, and the main results remain superficial and unexplored.

In more detail:

1. Abstract: The Abstract is confused, results should be presented briefly, and the Authors should provide at least two rows of background/introduction.

2. Materials and methods:

- Please specify why the Authors choose total testosterone instead free testosterone?

- Authors should provide more data about the causes of low testosterone levels, e.g., injury, trauma, infections, or tumors of the testes? Medications or radiations exposure? Liver diseases? Were these cases of hypogonadism primary or secondary? Did they perform any other tests to investigate the dysfunction of the pituitary gland? Etc. These data are essential to exclude the causality of presented results.

- What about smoking habits (that can influence both hypogonadism [David E. et al. JCEM, Volume 90, Issue 2, 1 February 2005, Pages 712–719] and HTGW [Gasevic, D. et al. Lipids Health Dis 13, 38 (2014).])?

- I recommend including all data relative to lipid profile for completeness.

3. Results: data are not clearly exposed, e.g., lines 119-120 they affirm that “there was a statistically significant difference between the phenotypes (NWNT, EW, HTG, and HTGW) and hypogonadism (p < 0.001)” – I suggest to re-phrase as “the percentage of EW, HTG, and HTGW was significantly higher in the group of patients with hypogonadism compared with those with normal testosterone values” and to add the exact p values for each group. In table 2 and table 3, what the p value referring to?

4. Discussion: I suggest including a more in-depth explanation of the possible pathogenic mechanisms involved in the relationship between metabolic disorders, such as HTGW, and low testosterone levels. Moreover, personally, I can hardly understand if hypogonadism is the consequence or a potential cause of HTGW.

6. PLOS authors have the option to publish the peer review history of their article (what does this mean?). If published, this will include your full peer review and any attached files.

Reviewer #1: No

Reviewer #2: No

---

## [Author Response · Author response to Decision Letter 0]

13 Jan 2022

Responses to reviewers’ comments on “Association between hypertriglyceridemic waist phenotype and hypogonadism in Taiwanese adult men”

(Submission ID: PONE-D-21-22955)

We appreciate the reviewers’ comments. Our point-by-point responses are provided below (Line and page numbers are shown for convenience, and the revised material is indicated in red text in the revised manuscript).

Reviewer #1:

Abstract:

lines 118-120 is not clear what the p value is referring to. In the text authors report that “there was a statistically significant difference between the phenotypes (NWNT, EW, HTG, and HTGW) and hypogonadism (p < 0.001).”, but from table 2 p value seems about the difference between group with and without hypogonadism. However, it appears from numbers displayed in table 2 that there is a higher prevalence of HTGW and a lower prevalence of NWNT in those with hypogonadism than in those without hypogonadism, but among those with hypogonadism it seems that the prevalence of HTGW is similar to that of NWNT. Nevertheless these are only impressions from numbers because pairwise p values are not reported. The same for the analyses showed in table 3 and lines 126-132.

Response: page 9, lines 131-134, 138-142

We thank the reviewer for their comment. With reference to the literature (Okada, 2020; Chien et al., 2020), we have adjusted the contents of Tables 2 and 3 to clearly express the correlation between phenotypes and hypogonadism. The p-value refers to the use of a chi-square test. We have modified the results section as follows:

The overall prevalence of hypogonadism among all participants was 10.63%. As shown in Table 2, the distribution of phenotypes (NWNT, EW, HTG, and HTGW) was significantly different between participants with and without hypogonadism (p <0.001). Participants without hypogonadism were more likely to have the NWNT phenotype than those with hypogonadism.

The prevalence of each phenotype in those with and without hypogonadism in each age group is shown in Table 3. In age group 1, participants with hypogonadism were significantly more likely to be HTGW than those without (p <0.001). In age group 2, the NWNT phenotype was significantly more prevalent among those with hypogonadism than those without (p <0.001). However, there were no significant differences in phenotypes between those with and without hypogonadism in age group 3.

References

Chien KH, Huang KH, Chung CH, Hsieh YH, Liang CM, Chang YH, et al. The impact of diabetes mellitus medication on the incidence of endogenous endophthalmitis. PLoS One 2020;15: e0227442.

Okada C, Takimoto H. (Development of a screening method for determining sodium intake based on the Dietary Reference Intakes for Japanese, 2020: a cross-sectional analysis of the National Health and Nutrition Survey, Japan. PLoS One 2020;15: e0235749.

1. Authors derives from chi-square test the correlation between these subgroups, but chi-square actually report information about the difference in distribution and not about correlation. Consequently, the logistic regression analyses is not really supported from previous analyses and implies a causal relationship that is not justified by the presented data.

I suggest analysing the variables as continuous variables and using appropriate tests to highlight possible correlations and then detect the presence of potential hypogonadism independent determinants through multivariate regression analysis.

Response: page 11, lines 147-155

We thank the reviewer for their comment. We conducted a linear regression analysis, the results of which are shown in the table below (table not in manuscript). The effects of phenotypes on hypogonadism were found to be similar using this method. We analyzed hypogonadism as a binary dependent variable according to the approach established in the prevalence of hypogonadism study by Mulligan et al. (2006). However, the total testosterone level distribution was found to be abnormal. Therefore, we used the results of the logistic regression model.

References

Mulligan T., Frick MF, Zuraw QC, Stemhagen A, & McWhirter C. Prevalence of hypogonadism in males aged at least 45 years: The HIM study. Int J Clin Pract 2006;60: 762-769.

Table 0. Results of linear regression analysis

 Age < 50 years

 (n = 4135, age group 1) Age 50–64 years 

(n = 1958, age group 2) Age ≥ 65 years

 (n = 349, age group 3)

 β p 95% CI β p 95% CI β p 95% CI

crude 

NWNT Ref. Ref. Ref. 

EW -121.09 <0.001*** -139.60, -105.59 -133.49 <0.001*** -161.71, -105.28 -126.84 <0.001*** -191.30, -62.39

HTG -103.76 <0.001*** -118.99, -88.53 -121.59 <0.001*** -148.30, -94.88 -99.43 0.014* -178.36, -20.49

HTGW -172.82 <0.001*** -191.11, -154.53 -161.99 <0.001*** -193.51, -130.46 -170.59 <0.001*** -247.47, -93.71

adjusted 

NWNT Ref. Ref. Ref. 

EW -18.01 0.098 -39.37, 3.35 -39.10 0.025* -73.22, -4.98 -32.23 0.429 -112.30, 47.83

HTG -75.68 <0.001*** -92.59, -58.77 -95.89 <0.001*** -125.30, -66.47 -71.19 0.107 -157.84, 15.46

HTGW -62.16 <0.001*** -84.88, -39.43 -66.25 <0.001*** -104.72, -27.77 -96.32 0.046* -190.89, -1.75

CI, confidence interval; EW, enlarged waist and normal triglyceride levels; HTG, hypertriglyceridemia and normal waist; HTGW, Hypertriglyceridemia and waist circumference ≥90 cm; NWNT, normal waist and normal triglyceride levels.

Reviewer #2:

1. Abstract: The Abstract is confused, results should be presented briefly, and the Authors should provide at least two rows of background/introduction.

Response: page 2, lines 26-29 

We thank the reviewer for their comment. We have adopted the reviewer’s suggestion and added the following to the abstract.

The hypertriglyceridemic enlarged waist (HTGW) phenotype is a tool for predicting abnormalities of cardiovascular metabolism. However, the relationship between the HTGW phenotype and hypogonadism remains undetermined.

2. Materials and methods:

- Please specify why the Authors choose total testosterone instead free testosterone?

Response: page 13, lines 208-212

We thank the reviewer for their comment. We used the health examination database in this study, from which free testosterone data could not be obtained. We have added the following passage to the limitations subsection.

Second, we evaluated testosterone deficiency based on a single measurement of total testosterone level because the free testosterone data could not be obtained from the health examination database used in this study. We hope to conduct another retrospective study in the future to investigate the relationship between the HTGW phenotype and free testosterone.

3. - Authors should provide more data about the causes of low testosterone levels, e.g., injury, trauma, infections, or tumors of the testes? Medications or radiations exposure? Liver diseases? Were these cases of hypogonadism primary or secondary? Did they perform any other tests to investigate the dysfunction of the pituitary gland? Etc. These data are essential to exclude the causality of presented results.

Response: Figure 2 and page 11, lines 147-155

We thank the reviewer for their comment. With reference to the literature (Laaksonen et al., 2005; Gasevic et al., 2014), we have included smoking status, cholesterol, HDL, and LDL in our analysis, and have revised the “Results” section as shown below. As the health database does not collect specific measures of nicotine intake, it was not possible to analyze the tobacco use dosages. 

As shown in Figure 2, we investigated the effects of the four phenotypes on the risk of hypogonadism. In age group 1, participants with the HTGW (odds ratio [OR], 1.980; 95% confidence interval [CI], 1.354-2.896) and HTG phenotype (OR, 1.803; 95% CI, 1.297-2.505) had higher risks of hypogonadism than those with the NWNT phenotype, after adjustment for BMI and FBG levels. In age group 2, participants with the HTGW (OR, 1.873; 95% CI, 1.099-3.193) and HTG (OR, 2.098; 95% CI, 1.334-3.298) phenotypes also had higher risks of hypogonadism than those with NWNT after adjustment for BMI, FBG level, cholesterol level, HDL level, LDL level, and smoking status. However, no relationship was observed between the phenotypes and hypogonadism in age group 3.

Odds ratios for hypogonadism in age group 1 (aged <50 years)

Odds ratios for hypogonadism in age group 2 (aged 50-64 years)

Odds ratios for hypogonadism in age group 3 (aged ≧65 years)

Figure 2. Odds ratio for hypogonadism stratified by age group. 

Adjusted OR were adjusted for body mass index, fasting blood glucose levels, cholesterol levels, high-density lipoprotein levels, low-density lipoprotein levels, and smoking status.

NWNT, normal waist circumference and normal triglyceride levels; HTG, hypertriglycer-idemia and normal waist circumference; EW, enlarged waist and normal triglyceride lev-els; HTGW, hypertriglyceridemia and waist circumference ≥90 cm.

References

Gasevic D, Carlsson AC, Lesser IA, et al. The association between “hypertriglyceridemic waist” and sub-clinical atherosclerosis in a multiethnic population: A cross-sectional study. Lipids Health Dis 2014;13: 38.

Laaksonen DE, Niskanen L, Punnonen K, Nyyssönen K, Tuomainen TP, Valkonen, VP, Salonen JT. The metabolic syndrome and smoking in relation to hypogonadism in middle-aged men: A prospective cohort study. J Clin Endocr Metab. 2005;90: 712-719.

4. Results: data are not clearly exposed, e.g., lines 119-120 they affirm that “there was a statistically significant difference between the phenotypes (NWNT, EW, HTG, and HTGW) and hypogonadism (p < 0.001)” – I suggest to re-phrase as “the percentage of EW, HTG, and HTGW was significantly higher in the group of patients with hypogonadism compared with those with normal testosterone values” and to add the exact p values for each group. In table 2 and table 3, what the p value referring to?

Response: page 9, lines 131-134, 138-142

We thank the reviewer for their comment. With reference to the literature (Chien et al., 2020; Okada & Takimoto, 2020), we adjusted Tables 2 and 3 to more clearly express the correlations between phenotypes and hypogonadism. The p-value refers to the use of the chi-square test. We have modified the results section as follows: 

The overall prevalence of hypogonadism in all participants was 10.63%. As shown in Table 2, the distribution of phenotypes (NWNT, EW, HTG, and HTGW) was significantly different between participants with and without hypogonadism (p <0.001). Participants without hypogonadism were more likely to have the NWNT phenotype than those with hypogonadism.

Table 3 shows the prevalence of hypogonadism in the different age groups. In age group 1, participants with hypogonadism were more likely to have the HTGW (p <0.001) phenotype. In age group 2, the NWNT phenotype was more prevalent among participants with hypogonadism (p <0.001). However, there was no significant relationship between hypogonadism and phenotypes in age group 3.

References

Chien KH, Huang KH, Chung CH, Hsieh YH, Liang CM, Chang YH, et al. The impact of diabetes mellitus medication on the incidence of endogenous endophthalmitis. PLoS One 2020;15: e0227442.

Okada C, Takimoto H. Development of a screening method for determining sodium intake based on the Dietary Reference Intakes for Japanese, 2020: A cross-sectional analysis of the National Health and Nutrition Survey, Japan. PLoS One 2020;15: e0235749.

4. Discussion: I suggest including a more in-depth explanation of the possible pathogenic mechanisms involved in the relationship between metabolic disorders, such as HTGW, and low testosterone levels. Moreover, personally, I can hardly understand if hypogonadism is the consequence or a potential cause of HTGW.

Response: page 12, lines 173-185

We thank the reviewer for their comment. We have adopted their suggestion and added the following passage to the discussion.

Several studies have shown that visceral obesity in men is associated with low testosterone levels, male infertility, and erectile dysfunction [24–28]. In animal experiments, weight loss was associated with a rise in levels of testosterone, free testosterone (FT), and sex hormone-binding globulin (SHBG), whereas weight gain was associated with a fall in these levels [29]. However, hypogonadism may induce visceral obesity. In a castrated animal model, weight significantly increased, especially in terms of visceral fat accumulation [30]. This implies that hypogonadism has a bidirectional relationship with obesity. Among the components of metabolic syndrome, it has the strongest association with visceral obesity [31, 32]. Indeed, male hypogonadism can result from complex interactions among lifestyle, metabolic health, and genetic background. Endocrine disruption caused by metabolic diseases can trigger the onset of hypogonadism, although the underlying mechanisms are not fully understood [33]. Metabolic disorders may modify the hypothalamic-pituitary-gonadal (HPA) axis by periodically suppressing one or more of its components; whereas, in permanent forms of hypogonadism, one or more of the HPA axis components irrevocably loses functionality [34].

References

24. Tchernof A, Després JP. Pathophysiology of human visceral obesity: An update. Physiol Rev. 2013;93: 359-404.

25. Amjad S, Baig M, Zahid N, Tariq S, Rehman R. Association between leptin, obesity, hormonal interplay, and male infertility. Andrologia 2019;51: e13147.

26. Huang IS, Mazur DJ, Kahn BE, Keeter MK, Desai AS, Lewis K, et al. Risk factors for hypogonadism in young men with erectile dysfunction. J Chin Med Assoc. 2019;82: 477-481.

27. Turan E, Öztekin Ü. Relationship between visceral adiposity index and male infertility. Andrologia 2020;52: e13548.

28. Sun K, Wang C, Lao G, Lin D, Huang C, Li N, et al. Lipid accumulation product and late-onset hypogonadism in middle-aged and elderly men: Results from a cross-sectional study in China. BMJ Open 2020;10: e033991.

29. Georgiev IP, Georgieva TM, Ivanov V, Dimitrova S, Kanelov I, Vlaykova T, et al. Effects of castration-induced visceral obesity and antioxidant treatment on lipid profile and insulin sensitivity in New Zealand white rabbits. Res Vet Sci. 2011;90: 196-204.

30. Camacho EM, Huhtaniemi IT, O'Neill TW, Finn JD, Pye SR, Lee DM, et al. Age-associated changes in hypothalamic-pituitary-testicular function in middle-aged and older men are modified by weight change and lifestyle factors: Longitudinal results from the European Male Ageing Study. Eur J Endocrinol. 2013;168: 445-55.

31. Crisóstomo L, Pereira SC, Monteiro MP, Raposo JF, Oliveira PF, Alves MG. Lifestyle, metabolic disorders and male hypogonadism – A one-way ticket? Mol Cell Endocrinol. 2020;516: 110945.

32. Brand JS, Rovers MM, Yeap BB, Schneider HJ, Tuomainen TP, Haring R, et al. Testos-terone, sex hormone-binding globulin, and the metabolic syndrome in men: An individual participant data meta-analysis of observational studies. PLoS One. 2014;9: e100409.

33. Wang N, Zhai H, Han B, Li Q, Chen Y, Chen Y, et al. Visceral fat dysfunction is positively associated with hypogonadism in Chinese men. Sci Rep. 2016;6: 19844.

34. Lunenfeld B, Mskhalaya G, Zitzmann M, Arver S, Kalinchenko S, Tishova Y, et al. Recom-mendations on the diagnosis, treatment, and monitoring of hypogonadism in men. The Ag-ing Male. 2015;18: 5-15.

---

## [Decision Letter · Decision Letter 1]

31 Jan 2022

PONE-D-21-22955R1Association between hypertriglyceridemic waist phenotype and hypogonadism in Taiwanese adult menPLOS ONE

Dear Dr. Chiu,

Thank you for submitting your manuscript to PLOS ONE. After careful consideration, we feel that it has merit but does not fully meet PLOS ONE’s publication criteria as it currently stands. Therefore, we invite you to submit a revised version of the manuscript that addresses the points raised during the review process.

We look forward to receiving your revised manuscript.

Kind regards,

Giacomo Pucci

Academic Editor

PLOS ONE

Journal Requirements:

Reviewers' comments:

Reviewer's Responses to Questions

**Comments to the Author**

1. If the authors have adequately addressed your comments raised in a previous round of review and you feel that this manuscript is now acceptable for publication, you may indicate that here to bypass the “Comments to the Author” section, enter your conflict of interest statement in the “Confidential to Editor” section, and submit your "Accept" recommendation.

Reviewer #1: All comments have been addressed

Reviewer #2: All comments have been addressed

2. Is the manuscript technically sound, and do the data support the conclusions?

Reviewer #1: Yes

Reviewer #2: Yes

3. Has the statistical analysis been performed appropriately and rigorously? 

Reviewer #1: Yes

Reviewer #2: Yes

4. Have the authors made all data underlying the findings in their manuscript fully available?

Reviewer #1: No

Reviewer #2: No

5. Is the manuscript presented in an intelligible fashion and written in standard English?

Reviewer #1: Yes

Reviewer #2: Yes

6. Review Comments to the Author

Reviewer #1: (No Response)

Reviewer #2: The new version of the manuscript seems to be now clearer and more comprehensible. In particular, the aim of the study is precise and the results are more intelligible. The Authors improved their text addressing to nearly all comments reported by Reviewers. I have one suggestion to improve even more the quality of this manuscript; in the Study Population section, I would specify that the population under the study was free from any pathological condition that affects the testes function (e.g. injury, trauma, infections, or tumors, endocrinological disorders, etc...).

7. PLOS authors have the option to publish the peer review history of their article (what does this mean?). If published, this will include your full peer review and any attached files.

Reviewer #1: No

Reviewer #2: No

---

## [Author Response · Author response to Decision Letter 1]

11 Feb 2022

Responses to reviewers’ comments on “Association between hypertriglyceridemic waist phenotype and hypogonadism in Taiwanese adult men”

(Submission ID: PONE-D-21-22955)

We appreciate the reviewers’ comments. Our point-by-point responses are provided below (Line and page numbers are shown for convenience, and the revised material is indicated in red text in the revised manuscript).

Reviewer #2:

The new version of the manuscript seems to be now clearer and more comprehensible. In particular, the aim of the study is precise and the results are more intelligible. The Authors improved their text addressing to nearly all comments reported by Reviewers. I have one suggestion to improve even more the quality of this manuscript; in the Study Population section, I would specify that the population under the study was free from any pathological condition that affects the testes function (e.g. injury, trauma, infections, or tumors, endocrinological disorders, etc...).

Response: page 4, lines 81-82

page 13, lines 211-216

We thank the reviewer for comment. We have modified the excluded criteria and added the following contents in the limitation subsection. 

Study population

Subsequently, we excluded participants with prostate cancer and insufficient data on serum testosterone level.

Limitations

This study has several limitations. First, this cross-sectional study could not clarify the causality between the HTGW phenotype and hypogonadism in the Asian population. Further studies with a prospective design are necessary to verify our findings in different ethnic populations. Second, we evaluated testosterone deficiency based on a single measurement of total testosterone level because the FT data could not be obtained from the health examination database used in this study. Furthermore, some pathological conditions that may affect testes function (eg injury, trauma, infection or tumor, endocrine disease, etc.) were not able to include in our data. We hope to conduct another retrospective study in the future to investigate the relationship between the HTGW phenotype and FT without previous pathological conditions. Third, this study did not use imaging technologies such as computed tomography and magnetic resonance imaging to evaluate the visceral fat content of the participants, so we could not understand the relationship between visceral fat and testosterone level. Additional studies are needed to examine this relationship.

---

## [Decision Letter · Decision Letter 2]

7 Mar 2022

Association between hypertriglyceridemic waist phenotype and hypogonadism in Taiwanese adult men

PONE-D-21-22955R2

Dear Dr. Chiu,

We’re pleased to inform you that your manuscript has been judged scientifically suitable for publication and will be formally accepted for publication once it meets all outstanding technical requirements.

Kind regards,

Giacomo Pucci

Academic Editor

PLOS ONE

Reviewer #2: All comments have been addressed

---

## [Editor Report · Acceptance letter]

15 Mar 2022

PONE-D-21-22955R2 

Association between hypertriglyceridemic waist phenotype and hypogonadism in Taiwanese adult men 

Dear Dr. Chiu:

I'm pleased to inform you that your manuscript has been deemed suitable for publication in PLOS ONE. Congratulations! Your manuscript is now with our production department. 

Kind regards, 

on behalf of

Dr. Giacomo Pucci 

Academic Editor

PLOS ONE